# Elevated Plasma Concentration of 4-Pyridone-3-carboxamide-1-β-D-ribonucleoside (4PYR) Highlights Malignancy of Renal Cell Carcinoma

**DOI:** 10.3390/ijms25042359

**Published:** 2024-02-17

**Authors:** Agata Jedrzejewska, Patrycja Jablonska, Teresa Gawlik-Jakubczak, Mateusz Czajkowski, Patrycja Maszka, Paulina Mierzejewska, Ryszard T. Smolenski, Ewa M. Slominska

**Affiliations:** 1Department of Biochemistry, Medical University of Gdansk, 80-211 Gdansk, Poland; agata.jedrzejewska@gumed.edu.pl (A.J.); patrycja.jablonska@gumed.edu.pl (P.J.); patrycja.maszka@gumed.edu.pl (P.M.); paulina.mierzejewska@gumed.edu.pl (P.M.); 2Department of Urology, Medical University of Gdansk, 80-211 Gdansk, Poland; teresaj@gumed.edu.pl (T.G.-J.); mateusz.czajkowski@gumed.edu.pl (M.C.)

**Keywords:** renal cell carcinoma, nicotinamide metabolism, 4-pyridone-3-carboxamide-1-β-D-ribonucleoside, prognosis, histopathological subtypes

## Abstract

Nicotinamide (NA) derivatives play crucial roles in various biological processes, such as inflammation, regulation of the cell cycle, and DNA repair. Recently, we proposed that 4-pyridone-3-carboxamide-1-β-D-ribonucleoside (4PYR), an unusual derivative of NA, could be classified as an oncometabolite in bladder, breast, and lung cancer. In this study, we investigated the relations between NA metabolism and the progression, recurrence, metastasis, and survival of patients diagnosed with different histological subtypes of renal cell carcinoma (RCC). We identified alterations in plasma NA metabolism, particularly in the clear cell RCC (ccRCC) subtype, compared to papillary RCC, chromophobe RCC, and oncocytoma. Patients with ccRCC also exhibited larger tumor sizes and elevated levels of diagnostic serum biomarkers, such as hsCRP concentration and ALP activity, which were positively correlated with the plasma 4PYR. Notably, 4PYR levels were elevated in advanced stages of ccRCC cancer and were associated with a highly aggressive phenotype of ccRCC. Additionally, elevated concentrations of 4PYR were related to a higher likelihood of mortality, recurrence, and particularly metastasis in ccRCC. These findings are consistent with other studies, suggesting that NA metabolism is accelerated in RCC, leading to abnormal concentrations of 4PYR. This supports the concept of 4PYR as an oncometabolite and a potential prognostic factor in the ccRCC subtype.

## 1. Introduction

Clear cell renal cell carcinoma (ccRCC) is the most prevalent type of kidney cancer, accounting for approximately 80% of all kidney cancer cases, and affecting men more than women [1]. It originates from the epithelial tissue of the proximal segments of renal tubules and mostly develops in the renal cortex, in the upper pole, surrounded by a connective tissue capsule. A much less frequently diagnosed kidney cancer is papillary renal cell carcinoma also known as chromophil (10–15% of cases) [2]. It is characterized by a papillary type of growth, often occurring bilaterally in the form of multiple foci. About 5% of cases are chromophobe renal cell carcinoma, which is named for the darker coloration of its cells [3]. Lastly, oncocytoma—a benign renal tumor—constitutes only 3–7% of renal tumors [4].

The cancer environment is also characterized by the presence of various inflammatory cells that produce cytokines, chemokines, and growth factors, involved in neoangiogenesis and metastases formation. These inflammatory mediators modulate the expression of genes important for cancer development and activate, for instance, NF-κB-dependent signaling pathways, which help the cell avoid apoptosis [5]. In this manner, oxidative stress is also unregulated, which drives mutagenesis and carcinogenesis [6,7,8]. In addition to malignant progression, chronic inflammation may cause endothelial dysfunction and fibrosis leading to worsening organ function [9]. Moreover, a vast majority of RCC risk factors, like smoking, obesity, and hypertension, are direct consequences of chronic inflammation. Thus, the targeting of inflammation is believed to be a promising strategy for both the prevention and therapeutic intervention of most cancers [10].

Nicotinamide (NA) is the main NAD^+^ derivative known for its anti-inflammatory properties and regulation of the cell cycle and DNA repair [11]. NA can be formed from nicotinic acid and then transformed into N-methylnicotinamide (MNA) by the enzyme nicotinamide methyltransferase (NNMT). Under the activity of aldehyde oxidase, MNA is converted into either N-methyl-2-pyridone-5-carboxamide (Met2PY) or N-methyl-4-pyridone-3-carboxamide (Met4PY), which are the final products of NA degradation [12].

Recently, the role of NNMT in cancer development has been extensively studied. It has been reported that an NNMT-encoding gene was overexpressed in several malignancies [13,14,15,16]. Moreover, NNMT downregulation was reported to reduce proliferation and tumor-forming in various cell lines [17], while enzyme overexpression resulted in enhanced cell viability [18]. Interestingly, NNMT gene expression was reported to be notably upregulated in the tumor tissue of RCC patients. NNMT expression was inversely correlated with tumor size [19,20], providing evidence for the theory that the enzyme may play a role in tumor growth [21,22]. Given the high mortality and poor prognosis in the case of metastases, Campagna et al. suggest that more research is urgently needed to identify new molecular targets in RCC, and NA metabolism could represent an interesting therapeutic approach [23].

Another much-less-known NA metabolic pathway leads to the formation of 4-pyridone-3-carboxamide-1-β-D-ribonucleoside (4PYR). 4PYR was first identified in 1979 in the urine of patients with chronic megaloblastic leukemia [24]. However, the exact compound structure was identified much later [25]. 4PYR has been included in the group of uremic toxins, as its level in the blood plasma of patients with chronic renal failure was significantly elevated [26]. Previously, we have reported a significant increase in the concentration of plasma 4PYR and its correlation with the advancement of bladder, breast, and lung cancer, which allowed us to define 4PYR as an oncometabolite [27,28,29]. Moreover, several lines of evidence suggest that 4PYR might modulate the invasive potential of tumor cells, their survival during dissemination, their adaptation during the colonization stage, and also participation in local inflammation, increasing the permeability of endothelial cells [29,30,31]. NA and MNA show anti-inflammatory properties but 4PYR is considered to be harmful to the endothelium [30,32]. In addition, it affects cell energy metabolism by inhibiting glycolysis and the AMPD pathway without changes in mitochondrial function [33].

Thus, the current study aimed to investigate the potential relationship between the nicotinamide derivatives in the plasma of renal cancer patients and disease occurrence and progression. In particular, this study evaluated the final metabolites considered to be uremic toxins (Met2PY and Met4PY, and 4PYR) and the neoteric oncometabolite (4PYR). Furthermore, we examined the utility of the 4PYR measurement for differential diagnosis, as well as the prognosis prediction of renal cancer.

## 2. Results

### 2.1. Characteristics of Renal Cell Carcinoma (RCC) Histopathologic Subtypes

The clinical characteristics of 65 RCC patients with a variety of histological subtypes: clear cell RCC (ccRCC), papillary and chromophobe RCC (P/Ch RCC), and oncocytoma, along with healthy controls are presented in Table 1. The vast majority represent the ccRCC type and regardless of the classification of kidney cancer, men outnumbered women. Based on the urea level, all of the RCC patients had preserved renal function. Among the comorbidities, the most common was hypertension, but it did not affect the level of NA metabolites in RCC patients (in Appendix A).

Furthermore, we analyzed prognostic factors in ccRCC, P/Ch RCC, and oncocytoma based on biochemical parameters and variations of the Glasgow Prognostic Score (GPS, mGPS, HS-mGPS). The GPS distribution in RCC subtypes is shown in Appendix A. A clinical biochemistry profile showed a significantly elevated level of high-sensitivity C-reactive protein (hsCRP) and alkaline phosphatase (ALP) activity alongside a higher score of mGPS in ccRCC patients compared to healthy controls. There were no significant differences in the level of diagnostic serum biomarkers in P/Ch RCC and oncocytoma in comparison to the control group.

Moreover, ccRCC patients had a significantly larger tumor size, compared to oncocytoma, which highlighted the malignancy of this cancer. In addition, patients with ccRCC might be characterized as more likely to metastasize (36.4% of patients), relapse (43.2% of patients), and have higher mortality (34.1% of patients), compared to P/Ch RCC (10%, 10%, and 0% of patients, respectively) and oncocytoma (0%, 12.5%, and 0% of patients, respectively). According to ccRCC malignancy, we also distinguished groups with different stages of cancer development (I, II, III, IV) and low- and high-aggressiveness types (low grade; stages I and II, and high grade; stages III and IV). The characteristics of these groups are shown in Appendix A.

### 2.2. Differential Plasma Nicotinamide Metabolic Pattern among RCC Histological Subtypes

To examine nicotinamide metabolic patterns in different histological subtypes, as well as in different stages of ccRCC, the concentration of NA, nicotinic acid, MNA, Met2PY, Met4PY, and 4PYR were measured in plasma. There were no significant differences in the NA derivates in P/Ch RCC patients compared to healthy controls. Interestingly, a significantly higher concentration of 4PYR was reported in ccRCC and oncocytoma patients, compared to the control group (Figure 1A). However, only a trend toward elevated levels of 4PYR with a higher stage of ccRCC was observed(Figure 1B). We also showed increased concentrations of Met2PY and Met4PY in ccRCC, which resulted from significantly higher levels of these metabolites in stage III. While nicotinic acid was notably elevated during cancer development (stage III, IV), the level of MNA significantly increased in stage II of ccRCC.

Then, we focused on the relationship between NA and its catabolites in the same types of renal cancer: ccRCC, P/Ch RCC, and oncocytoma. We observed a positive correlation between nicotinic acid and NA (Figure 2A,E,G) as well as between Met2PY and Met4PY (Figure 2B,F,H) in each of the three subtypes of renal cancer. Only in the plasma of ccRCC did 4PYR linearly correlate with Met2PY and Met4PY (Figure 2C,D).

### 2.3. Strong Association of 4PYR with the Clinical Factors of Progression and Highly Aggressive RCC Phenotype

In correlation analyses of NA metabolites with all prognostic factors, we observed that the plasma 4PYR level positively correlates with hsCRP and the size of the tumor (Table 2), and it was previously observed to be unfavorably increased in patients with ccRCC. We also noted a significantly positive correlation of 4PYR with all variants of the Glasgow Prognostic Score (GPS, mGPS, and HS-GPS) and a negative correlation with cholesterol. Additionally, ALP correlated negatively with NA and MNA, and albumin (ALB) with MNA, and urea correlated positively with Met2PY.

Our previous results have shown that NA metabolites, especially 4PYR, were significantly elevated in the plasma of ccRCC patients, which were also highly associated with factors of a worse prognosis. Hence, we aimed to confirm their increased presence in the high-malignancy ccRCC phenotype (high grade) compared to low-aggressiveness ccRCC (low grade) and healthy controls. Consistently, 4PYR exhibited the highest statistical elevation in high-grade ccRCC, followed by Met2PY (Figure 3A). Nicotinic acid and Met4PY were the least statistically significant between high-grade ccRCC and healthy controls. However, among NA metabolites, only 4PYR might distinguish patients with large tumor masses (>4 cm) from small tumor masses (Figure 3B).

Furthermore, we analyzed the correlation between plasma concentration of 4PYR and patient mortality, recurrence, and metastasis at the ROC analysis level (Figure 4A–C). Based on the AUC value, we observed a strong connection between 4PYR and worse prognosis, with the highest likelihood of separation in patients with and without metastases (Figure 4C).

## 3. Discussion

This study, for the first time, demonstrated the accelerated metabolism of nicotinamide (NA) in plasma, with particular emphasis on the final metabolites—Met2PY and Met4PY, but especially 4PYR, in different subtypes of renal cell carcinoma (RCC). In addition, its relationship to clinical parameters of the disease, such as tumor sizes, hsCRP concentration, ALP activity, and GPS score was shown. These data clearly highlight that 4PYR and other final NA metabolites could represent a potential biomarker of RCC.

In our study, ALP activity but not lactate dehydrogenase (LDH) was elevated in cancer patients compared to the control group. Higher levels of ALP have been linked to a less favorable outcome in several types of cancer, including colorectal, nasopharyngeal, prostate, and esophageal squamous cell carcinoma [34,35]. Our studies have shown that ALP correlates negatively with NAD^+^ salvage pathway metabolites, including NA and MNA, and positively with 4PYR. Furthermore, the level of hsCRP was significantly elevated in ccRCC and positively correlated with 4PYR concentration. Recently, CRP has been considered as a diagnostic and prognostic factor in the development of endometrial cancer [36]. From hsCRP and albumin measurements, we established the GPS score, an inflammation-based scale that has been reported to predict the biological behavior of malignancies [36,37]. Similarly, we observed a higher score of mGPS in patients with ccRCC that was also associated with elevated 4PYR concentration. Moreover, Met2PY and Met4PY positively correlated with urea levels. This is in line with recent results that these compounds are considered to be uremic toxins and the accumulation of Met2PY may directly contribute to uremia symptoms in patients with renal failure [38,39]. The negative action of Met2PY may also result from the inhibitory effect of poly(ADP-ribose)-1 polymerase (PARP-1) and increased sensitivity to DNA damage. One of the possible therapeutic approaches to lowering the disturbed concentration of uremic toxins is the use of NNMT inhibitors. Most often, these are small-molecule compounds, but in recent years, other bisubstrate inhibitors have also been discovered that have cellular activity and effectively reduce NMMT activity [23,40,41,42]. Therefore, several studies have already proposed the inhibition of NNMT as a new therapeutic strategy in oncology [40,43]. It has been shown that bisubstrate NNMT inhibitors significantly lower the concentration of MNA in lung carcinoma cells and endothelial cells [40,41]. In turn, treatment with small interfering RNAs (siRNAs) targeting NNMT efficiently suppressed the proliferation and invasive capacity of RCC cells [44].

In our research, we observed significant increases in the plasma concentrations of 4PYR, Met2PY, and Met4PY, along with the up-regulated formation of other NAD^+^ derivatives in ccRCC. Interestingly, nicotinic acid was significantly elevated in stages III and IV, while Met2PY and Met4PY might differentiate stage III with MNA, serving as a predictor of stage II. 4PYR was remarkably increased in the plasma of patients with ccRCC with a high-level tendency during cancer development. Concomitantly, patients with ccRCC with a larger tumor size and more aggressive tumor phenotype were characterized by elevated levels of 4PYR, which may indicate significant participation of this compound in the cancer progression. However, the main limitation of our study was the relatively small cohort of patients with ccRCC, particularly in the papillary/chromophobe RCC and oncocytoma groups. Additionally, if gene expression analyses, especially of NNMT, had been conducted in tumor tissue, this could have reinforced the significance of NA metabolism in RCC and provided insights into its impact on the level of 4PYR or other uremic toxins. In addition, the measured compounds have low tissue specificity and both 4PYR and the activity of NNMT are present in many tissues. Therefore, the analysis of comorbidities should be conducted reliably. Nevertheless, in our study, the level of 4PYR positively correlated with tumor masses and inflammatory parameters.

In conclusion, NA metabolism is altered in the course of ccRCC, and the measurement of 4PYR concentration together with inflammatory factors may be useful in clinical screening and diagnosis. Based on the performed ROC analysis, it can be assumed that measurements of plasma 4PYR concentration may be useful as a predictor of a patient’s survival, recurrence, and especially metastasis of ccRCC. Further research is needed to better link measurable blood parameters with the diagnosis and subsequent treatment of cancer patients.

## 4. Materials and Methods

### 4.1. Patients

The study was performed based on the standards of the Declaration of Helsinki and it was approved by the Independent Bioethics Committee for Scientific Research at the Medical University of Gdansk, Poland (protocol code: NKBBN/539/2014-2015). Informed consent has been obtained from the patients in their referring clinical center (Department of Urology at the Medical University of Gdansk).

Participants selected for the control group underwent elective treatment in the Department of Urology for benign urological disorders, including benign prostatic hyperplasia, urolithiasis, and stress urinary incontinence. All participants in the control group were in good overall health, without a history of cancer treatment or severe circulatory and respiratory conditions.

The inclusion criteria for this study were based on confirmation of a renal tumor, as determined by abdominal computed tomography, and eligibility for either nephrectomy or nephron-sparing surgery. The exclusion criteria were age below 18 years, the presence of any other type of cancer, multifocal and/or bilateral kidney tumors, and Von Hippel-Lindau disease. Plasma from the peripheral blood was collected shortly before surgery at fasting state into a sodium citrate tube, while serum was obtained from clotted blood without the addition of an anticoagulant. Both plasma and serum were acquired after centrifugation at 1200× *g* for 5 min at 21 °C. During surgery, a representative tumor sample was immediately collected for histopathological examination. The classification into appropriate tumor categories was performed according to the WHO classification system (2004) [4]. The sampling method is shown in Figure 5.

### 4.2. Determination of Patients’ Plasma Nicotinamide Metabolites

The concentration of nicotinamide metabolites was measured based on the previous protocol [45]. Briefly, 25 μL of plasma with 5 μL of the internal standard (2-chloradenosine) was extracted with 70 μL of acetonitrile. After 20 min of incubation on ice, samples were centrifuged at 16,000× *g* (4 °C, 10 min) and the supernatant was evaporated. The obtained precipitate was dissolved in water at a volume equal to the initial plasma volume and extracted with methanol (ratio 1:3 *v*/*v*). Then, samples were successively incubated, centrifuged, and evaporated as above. The final residue was reconstituted in 25 μL of water and analyzed by combined liquid chromatography/mass spectrometry (LC/MS).

### 4.3. Assessment of Routine Prognostic Biochemical Parameters

The serum or plasma of patients was analyzed for several laboratory parameters, including ALB, ALP, hsCRP, calcium, LDH, urea, and cholesterol, using an automated photometer (ERBA XL-180, Mannheim, Germany) and specific ERBA kits. The measurements were performed following the protocols specified by the manufacturer. The Glasgow Prognostic Score (GPS), its modified version (mGPS), and the high-sensitivity mGPS (HS-mGPS) were assessed based on the Hirahara et al. protocol [46]. GPS was calculated by allocating one point each for elevated hsCRP (greater than 1.0 mg/dL) and hypoalbuminemia (albumin levels less than 3.5 mg/dL), while patients with either elevated hsCRP or hypoalbuminemia received only 1 point and patients with neither received 0. The mGPS assigns a score of 2 to patients with increased hsCRP levels (above 1.0 mg/dL) and reduced albumin (below 3.5 mg/dL), a score of 1 to those with increased hsCRP levels alone, and a score of 0 to those with normal hsCRP, regardless of albumin levels. On the contrary, the HS-mGPS uses different cut-off values for hsCRP level. Patients with heightened hsCRP levels (above 0.3 mg/dL) and hypoalbuminemia (below 3.5 mg/dL) received a score of 2, those with high hsCRP levels alone received a score of 1, and 0 for patients with a normal hsCRP, with or without hypoalbuminemia.

### 4.4. Statistical Analysis

The statistical analyses were conducted using InStat software (GraphPad Prism, version 8.0.1, San Diego, CA, USA) and STATISTICA software (version 13.0.) from www.statsoft.com. Normality was assessed using the Shapiro–Wilk normality tests. Comparisons between groups were evaluated after assessment of normality by one-way analysis of variance (ANOVA) followed by Holm–Sidak post-hoc test or Kruskal–Wallis test (Dunn’s post-hoc test). Similarly, an unpaired Student’s *t*-test or Mann–Whitney U test was used to distinguish the two groups. In the correlation analysis, the Pearson correlation coefficient was used. A *p*-value > 0.05 indicates the statistical significance, and the standard error of the mean (SEM) is represented by error bars in the figures. Receiver operating characteristic (ROC) analysis was performed for 4PYR associated with a prognostic value (survival/recurrence/metastatic). The best cut-off was determined using the Youden index and area under the curve (AUC).

## 5. Conclusions

In summary, for the first time, our study demonstrated the alterations in NA metabolism in RCC highlighting potential of its final derivatives, particularly 4PYR, as prognostic biomarkers in the ccRCC subtype. Disturbances in NA metabolism and the appearance of the end products of its metabolism were associated with ccRCC development and progression. High concentrations of 4PYR in the plasma of patients with ccRCC were positively correlated with increased levels of hsCRP and ALP activity. Higher 4PYR concentrations were associated with advanced clinical stages of ccRCC but also poorer prognoses in terms of tumor size, survival, recurrence, and metastases. Therefore, we indicated that 4PYR might be an oncometabolite and prognostic factor in ccRCC.

## Figures and Tables

**Figure 1 ijms-25-02359-f001:**
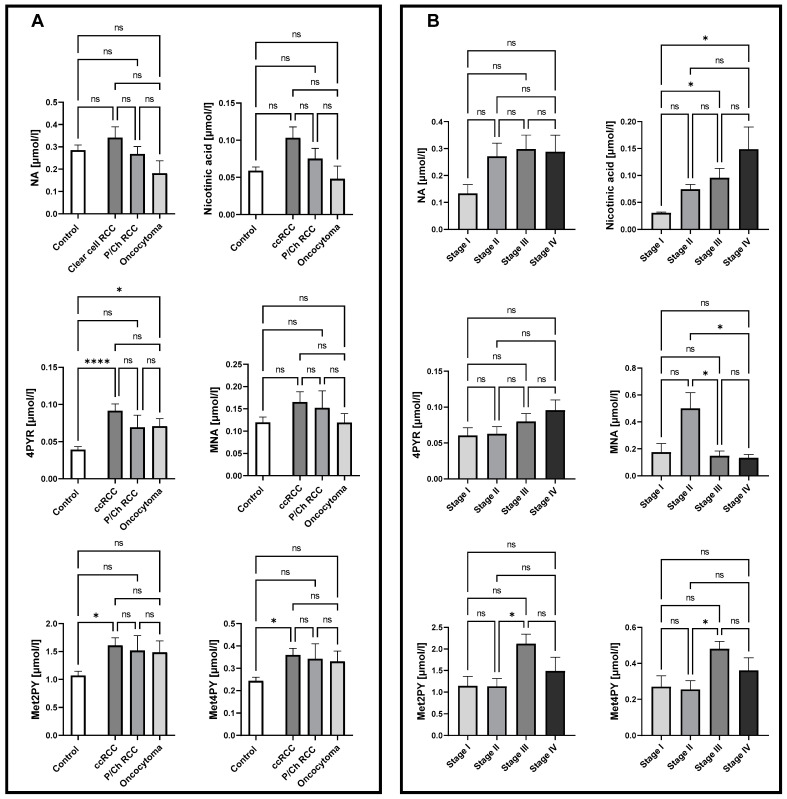
Significantly elevated levels of 4PYR in both clear cell renal cell carcinoma (ccRCC; *n* = 47) and oncocytoma (*n* = 8) alongside Met2PY and Met4PY up-regulation in ccRCC, with no changes in papillary and chromophobe RCC (P/Ch RCC *n* = 10) NA pattern, compared to healthy controls (*n* = 50) (**A**). Complex alterations in NA catabolites allow us to distinguish patients with stage I from stage III and IV ccRCC (*n* = 5/16/13) based on nicotinic acid level, stage II (*n* = 10) from III and IV ccRCC through MNA concentration, and stage II from III ccRCC via Met2PY and Met4PY levels (**B**). Results are shown as mean ± SEM, ns – not significant, * *p* < 0.05, and **** *p* < 0.001 by one-way ANOVA followed by Kruskal–Wallis test and Dunn’s post-hoc test.

**Figure 2 ijms-25-02359-f002:**
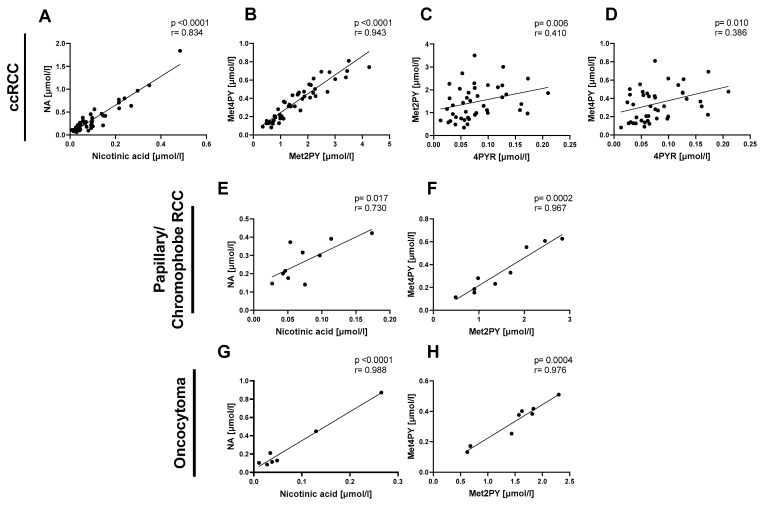
The significant relationship between nicotinamide-related metabolites in patients with clear cell renal cell carcinoma (ccRCC *n* = 47), papillary/chromophobe RCC (*n* = 10), and oncocytoma (*n* = 8). Correlation plots of plasma nicotinic acid with NA, Met2PY with Met4PY, and 4PYR with Met2PY and Met4PY in ccRCC (**A**–**D**), nicotinic acid with NA and Met2PY with Met4PY in papillary and chromophobe RCC (**E**,**F**), and nicotinic acid with NA, and Met2PY with Met4PY in oncocytoma (**G**,**H**). Results are shown as plots of the Pearson correlation coefficient (*r*) with the corresponding *p*-value (*p*).

**Figure 3 ijms-25-02359-f003:**
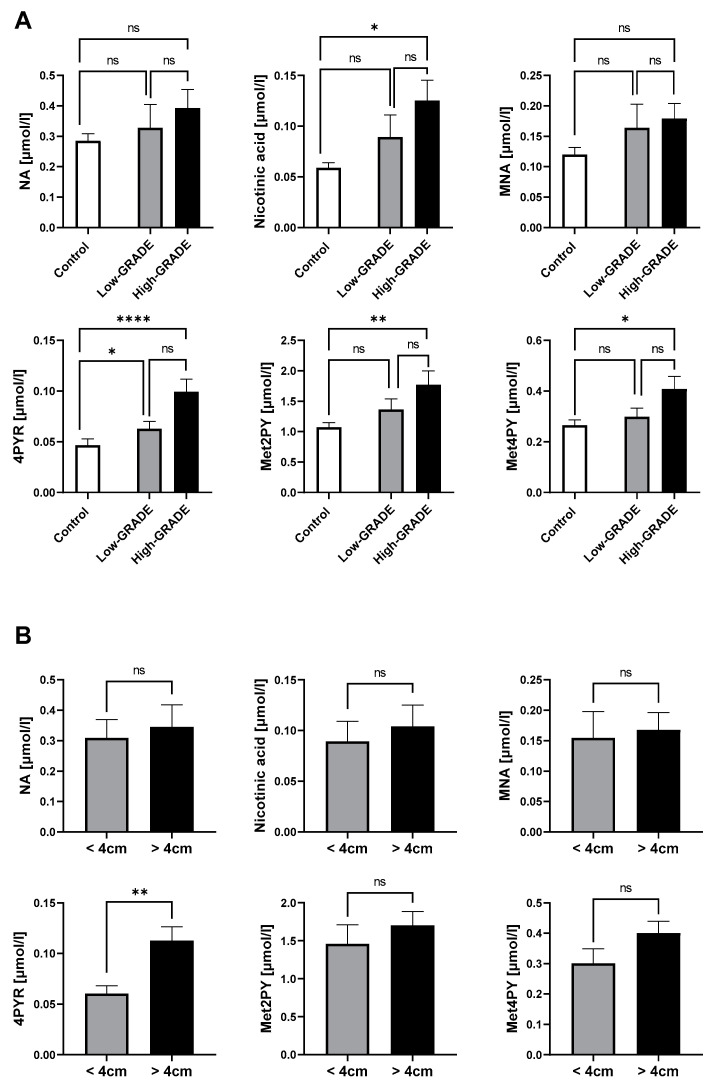
Elevated 4PYR levels prevail in clear cell RCC with a worse prognosis (*n* = 19), compared to the less invasive type (*n* = 25) and healthy controls (*n =* 50) (**A**), and among patients with a larger tumor mass (*n* = 27) than small tumor mass (*n* = 16) (**B**). Results are mean ± SEM, ns—not significant, * *p* < 0.05, ** *p* < 0.01, and **** *p* < 0.001 by Kruskal–Wallis test followed by Dunn’s post-hoc test (**A**) and Mann–Whitney U test (**B**).

**Figure 4 ijms-25-02359-f004:**
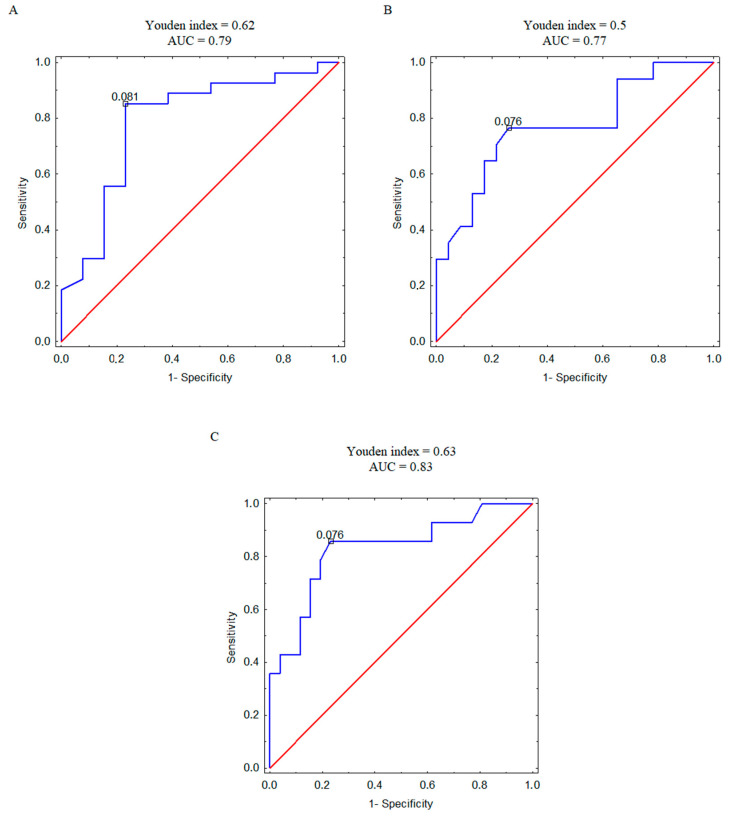
ROC curves with the corresponding values of Youden index and area under the curve (AUC) for 4PYR levels discriminating ccRCC survivors and non-survivors (**A**), relapsed and non-relapsed patients (**B**), and metastatic and non-metastatic patients (**C**). Blue line—ROC curve for ccRCC patients, red line—reference line.

**Figure 5 ijms-25-02359-f005:**
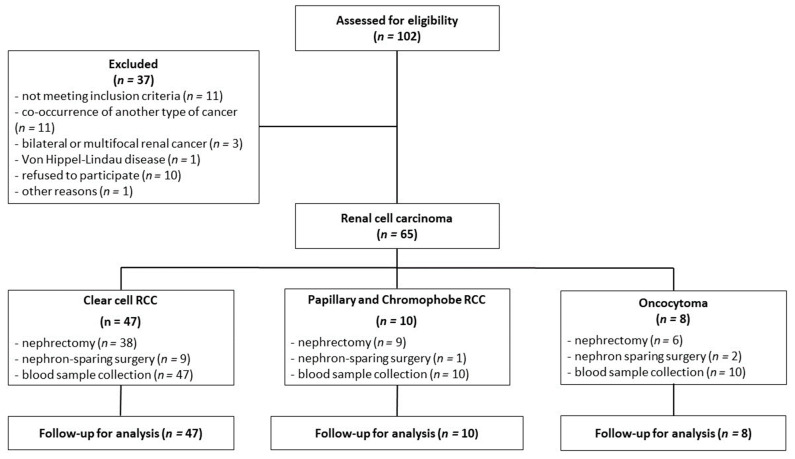
The flowchart of renal cell carcinoma patients.

**Table 1 ijms-25-02359-t001:** Clinical characteristics with biochemical profile of renal cell carcinoma histological subgroups and healthy controls. Results are shown as mean ± SEM, * *p* < 0.05, ** *p* < 0.01 vs. control, $ *p* > 0.05 vs. P/Ch RCC, and # *p* < 0.05 vs. renal oncocytoma by one-way ANOVA followed by Kruskal–Wallis test and Dunn’s post-hoc test. ccRCC, clear cell renal cell carcinoma; P/Ch RCC, papillary and chromophobe RCC; hsCRP, high-sensitivity C-reactive protein; CHOL, total cholesterol; ALB, albumin; ALP, alkaline phosphatase; Ca, calcium; LDH, lactate dehydrogenase; GPS, Glasgow Prognostic Score; mGPS modified Glasgow Prognostic Score; HS-mGPS, high-sensitivity modified Glasgow Prognostic Score; na., not applicable; n, number.

Parameter	Control *n =* 50	ccRCC *n =* 47	P/Ch RCC *n =* 10	Oncocytoma *n* = 8
Sex ratio (F/M)	11/39	16/31	4/6	2/6
Age, years	56 ± 3	63 ± 2	57 ± 5	72 ± 4 *
Median age, years	62	67	63	74
Comorbidity (n)				
Hypertension	21	13	0	3
Diabetes mellitus	11	6	0	2
Dyslipidemia	10	2	0	1
Ischemic heart disease	4	2	0	0
Benign prostatic hyperplasia	14	3	0	2
Rheumatoid arthritis	0	1	0	0
Biochemical data				
hsCRP, mg/dL	0.37 ± 0.05	2.69 ± 0.82 **	0.31 ± 0.08	0.4 ± 0.12
CHOL, mg/dL	148 ± 4.65	142 ± 6.67	146 ± 12.10	139 ± 7.78
ALB, g/dL	3.54 ± 0.04	3.4 ± 0.08	3.62 ± 0.10	3.61 ± 0.10
ALP, U/L	108 ± 4.88	145 ± 12.90 *	111 ± 12.20	123.5 ± 14.40
Ca, mg/dL	6.94 ± 0.06	7.18 ± 0.11	7.01 ± 0.24	7.07 ± 0.13
LDH, U/L	310 ± 12.20	317 ± 11.60	337 ± 25.80	342 ± 26.70
Urea, mg/dL	34 ± 1.29	34.2 ± 1.69	40.7 ± 7.50	31.4 ± 2.35
GPS score	0.65 ± 0.09	0.57 ± 0.13	0.2 ± 0.2	0.13 ± 0.13
mGPS score	0.19 ± 0.08	0.57 ± 0.13 *	0.2 ± 0.2	0.13 ± 0.13
HS-GPS score	0.58 ± 0.12	0.89 ± 0.13 $	0.2 ± 0.2	0.63 ± 0.26
Neoplasia characteristics				
Tumor size, cm	na.	6.16 ± 0.59 #	3.49 ± 0.6	2.38 ± 0.24
Nephrectomy (no/yes)	na.	9/38	1/9	2/6
Metastatic (no/yes)	na.	28/16	9/1	6/0
Recurrence (no/yes)	na.	25/19	9/1	7/1
Survival (no/yes)	na.	15/29	0/10	0/7

**Table 2 ijms-25-02359-t002:** Prognostic factor correlations with the concentration of plasma nicotinamide derivatives in ccRCC patients. Results are shown as Pearson correlation coefficient value (*r*) with corresponding *p*-value (*p*), ns – not significant, * *p* < 0.05, ** *p* < 0.01, and **** *p* < 0.001. NA, nicotinamide; MNA, N-methylnicotinamide; Met2PY, N-methyl-2-pyridone-5-carboxamide; Met4PY, N-methyl-4-pyridone-3-carboxamide; 4PYR, 4-pyridone-3-carboxamide-1-β-D-ribonucleoside; hsCRP, high-sensitivity C-reactive protein; CHOL, total cholesterol; ALB, albumin; ALP, alkaline phosphatase; Ca, calcium; LDH, lactate dehydrogenase; GPS, Glasgow Prognostic Score; mGPS modified Glasgow Prognostic Score; HS-mGPS, high-sensitivity modified Glasgow Prognostic Score.

Parameters	NA		Nicotinic Acid		MNA		4PYR		Met2PY		Met4PY	
	*r*	*p*	*r*	*p*	*r*	*p*	*r*	*p*	*r*	*p*	*r*	*p*
hsCRP, mg/dL	−0.04	ns	−0.11	ns	−0.09	ns	0.35	*	0.22	ns	−0.20	ns
CHOL, mg/dL	−0.06	ns	−0.08	ns	−0.22	ns	−0.35	*	−0.18	ns	0.10	ns
ALB, g/dL	0.05	ns	0.1	ns	−0.32	*	−0.29	ns	−0.03	ns	0.16	ns
ALP, U/L	−0.32	*	−0.24	ns	−0.40	*	0.30	ns	0.25	ns	0.04	ns
Ca, mg/dL	0.05	ns	0.02	ns	−0.24	ns	0.11	ns	0.22	ns	0.03	ns
LDH, U/L	0.11	ns	0.20	ns	0.04	ns	0.14	ns	−0.04	ns	0.09	ns
Urea, mg/dL	0.16	ns	0.17	ns	−0.05	ns	0.16	ns	0.71	****	0.15	ns
Tumor size, cm	−0.08	ns	−0.04	ns	0.04	ns	0.45	**	0.19	ns	0.05	ns
GPS score	−0.01	ns	0.01	ns	−0.01	ns	0.33	*	0.15	ns	−0.06	ns
mGPS score	−0.09	ns	−0.01	ns	−0.03	ns	0.35	*	0.18	ns	0.02	ns
HS-mGPS score	−0.03	ns	−0.04	ns	−0.07	ns	0.33	*	0.26	ns	−0.09	ns

## Data Availability

The data presented in this study are available on request from the corresponding authors.

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
