# Peer review of "Elevated Plasma Concentration of 4-Pyridone-3-carboxamide-1-β-D-ribonucleoside (4PYR) Highlights Malignancy of Renal Cell Carcinoma"

_ijms, 2024, doi:10.3390/ijms25042359_

Round 1

Reviewer 1 Report

Comments and Suggestions for Authors

The article “Elevated plasma concentration of 4-pyridone-3-carboxamide ribonucleoside (4PYR) highlights malignancy of Renal Cell Carcinoma” is a study regarding the NA metabolites in RCC and their possible role as prognostic factors in ccRCC subtypes, in particular 4PYR.

The manuscript is itself interesting and might be of interest for the readers. However, the manuscript contains several flaws that prevent its publication, thus authors are strongly encouraged to address the following concerns:

1.       Authors are encouraged to perform an extensive English revision of the manuscript.

2.       Authors should write NAD with “+” in uppercase.

3.       The introduction section does not provide any information about nicotinamide N-methyltransferase, the enzyme responsible of generating methylnicotinamide which is further transformed in 4PYR. That is really weird, since it has been demonstrated long time ago that NNMT is upregulated in ccRCC ans correlated with the prognosis, as recently reviewed (PMID: 34439880). Authors write a paragraph “Transformations activity of Nicotinamide Derivatives Suggests Increased Metabolism of Nicotinamide in RCC” but the fact that the enzyme is overexpressed and thus the pathway is boosted in ccRCC this has been demonstrated long time ago.

4.       How authors recruited the healthy controls? Did they check if they were healthy or affected by any other pathology or under treatment?

5.       How have you ended up with the sample of ccRCC subtypes? How many specimens were selected? How many were excluded? What were the reasons for exclusions? Please show a flowchart describing the sampling method.

6.       The sample was not matched by age. Could it be a bias for the interpretation of the results?

7.       In table 1 it is not clear authors should specify that they report mean age, and they must also show median age.

8.       The quality of figure 1 is so poor that is almost impossible to read it.

9.       Due to the high levels of 4PYR that correlates with the aggressiveness of RCC, it is conceivable to propose the use of NNMT inhibitors for clinical applications. Notably, a number of NNMT inhibitors are already available and were proposed as a promising strategy for cancer treatment (PMID: 34572571; PMID: 34704059; PMID: 34424711). Please discuss.

10.   In a total 27 references, 7 are of the authors (25%), which seemed excessive but not inappropriate.

Comments on the Quality of English Language

Moderate editing of English language required

Author Response

Response to the Reviewers’ comments

“Elevated plasma concentration of 4-pyridone-3-carboxamide ribonucleoside (4PYR) highlights malignancy of Renal Cell Carcinoma”

Manuscript ID: ijms-2837156

The article “Elevated plasma concentration of 4-pyridone-3-carboxamide ribonucleoside (4PYR) highlights malignancy of Renal Cell Carcinoma” is a study regarding the NA metabolites in RCC and their possible role as prognostic factors in ccRCC subtypes, in particular 4PYR.

The manuscript is itself interesting and might be of interest for the readers. However, the manuscript contains several flaws that prevent its publication, thus authors are strongly encouraged to address the following concerns:

  1. Authors are encouraged to perform an extensive English revision of the manuscript.

Thank you for the thorough review of our manuscript. We have revised this manuscript and corrected the English language.

  1. Authors should write NAD with “+” in uppercase.

       As suggested, we have changed NAD+ to NAD+

  1. The introduction section does not provide any information about nicotinamide N-methyltransferase, the enzyme responsible of generating methylnicotinamide which is further transformed in 4PYR. That is really weird, since it has been demonstrated long time ago that NNMT is upregulated in ccRCC ans correlated with the prognosis, as recently reviewed (PMID: 34439880). Authors write a paragraph “Transformations activity of Nicotinamide Derivatives Suggests Increased Metabolism of Nicotinamide in RCC” but the fact that the enzyme is overexpressed and thus the pathway is boosted in ccRCC this has been demonstrated long time ago.

Thanks to the Reviewer's comments we have provided more information about NNMT expression in RCC and its correlation with prognosis in the introduction section (l. 61-71). Therefore, we have rephrased the paragraph “Transformations activity of Nicotinamide Derivatives Suggests Increased Metabolism of Nicotinamide in RCC” as suggested (I. 128)

  1. How authors recruited the healthy controls? Did they check if they were healthy or affected by any other pathology or under treatment?

      Information about the recruitment of healthy individuals has been added to the methodology section (l.267-271). We clarified that all participants in the control group were in good overall health and added information about eventual comorbidity to Table 1.

  1. How have you ended up with the sample of ccRCC subtypes? How many specimens were selected? How many were excluded? What were the reasons for exclusions? Please show a flowchart describing the sampling method.

      As the Reviewer suggested, we have added a flowchart describing the sampling method (Figure 5.).

  1. The sample was not matched by age. Could it be a bias for the interpretation of the results?

In our analysis, there is no difference in age between the healthy control and the ccRCC patients (p= 0.14), a group that constitutes the vast majority of all patients. Therefore, age should not affect on results presented in the manuscript. After one-way ANOVA followed by Kruskal-Wallis test, the age difference occurs only between the control group and the Oncocytoma group, which constitutes a small percentage of all patients. The results were added in Table 1.

  1. In table 1 it is not clear authors should specify that they report mean age, and they must also show median age.

We have better defined the age of patients and controls in Table 1. Now, age is also presented with the median.

  1. The quality of figure 1 is so poor that is almost impossible to read it.

Figure 1. has been added with better quality.  

  1. Due to the high levels of 4PYR that correlates with the aggressiveness of RCC, it is conceivable to propose the use of NNMT inhibitors for clinical applications. Notably, a number of NNMT inhibitors are already available and were proposed as a promising strategy for cancer treatment (PMID: 34572571; PMID: 34704059; PMID: 34424711). Please discuss.

We have expanded the discussion section with information about the use of NNMT inhibitors in oncology treatment (l. 227-235).

  1. In a total 27 references, 7 are of the authors (25%), which seemed excessive but not inappropriate.

We added more non-self-citations due to changes in the introduction and discussion sections. Now, self-citations are below 15%.

Reviewer 2 Report

Comments and Suggestions for Authors

Dear Authors, I have read with interest your manuscript. My opinion is that this topic is very interesting to discuss, because nowdays we are confronting with a large burden of neoplasia patients and we need a better assessmnent of this condition.

I would like to address a few suggestions/ questions:

I think the title is very suggestive of your study and it is properly chosen. My opinion is that you should talk more in introduction about inflammation and its implications in neoplasia development, because this is the main subject of your study and also please bring more motivation of why you decided to write this study. NA metabolites are increased both in conditions like chronic kidney disease and also cancer, like you clearly stated from medical literature evidence, but how can we differentiate from this conditions and which are the cutoff values available in this moment in practice?  You clearly state the exclusion criteria, but you could also state de inclusion criteria?

My opinion is that you should have brought more details about neoplasia history of each patients and about its associated comorbidities. In this matter, you should include a complete medical history, and then perform multivariate analysis between different conditions, neoplasia type and NA metabolites values.

In your discussion section, you should talk more about the results from your study and not to include so much evidence from current medical literature and studies already published regarding this fields. I think these details should we written in introduction section, were you should explain more about the current evidence regarding this topic, bringing the latest highlights in this topic; thereafter, in your results focus only on your results and explain the best results from your study, and then explain what your study brings new in this field.

In conclusion, please state only the novelty of your study, without pulling conclusion from current literature, because this section of your paper should be based only on paper originality. Also, you lack the limitations of your study and we can identify some majore ones like incomplete medical history of the patients and associated comorbidites, small cohort of patients. Please explain is your study brings something new in this field and how you can integrate your results into clinical practice, how your study results could be included maybe in prognostic scores or diagnostic tools.

This topic is very interesting to discuss because we need a better management of neoplasia patients, given the fact that this is a condition with increased morbidity and mortality. We need better ways to asses this patients to increase treatment success.

Author Response

Response to the Reviewers’ comments

“Elevated plasma concentration of 4-pyridone-3-carboxamide ribonucleoside (4PYR) highlights malignancy of Renal Cell Carcinoma”

Manuscript ID: ijms-2837156

Dear Authors, I have read with interest your manuscript. My opinion is that this topic is very interesting to discuss, because nowdays we are confronting with a large burden of neoplasia patients and we need a better assessmnent of this condition.

Thank you for reviewing our work and allowing it to be published.

I would like to address a few suggestions/ questions:

I think the title is very suggestive of your study and it is properly chosen. My opinion is that you should talk more in introduction about inflammation and its implications in neoplasia development, because this is the main subject of your study and also please bring more motivation of why you decided to write this study. NA metabolites are increased both in conditions like chronic kidney disease and also cancer, like you clearly stated from medical literature evidence, but how can we differentiate from this conditions and which are the cutoff values available in this moment in practice?  You clearly state the exclusion criteria, but you could also state de inclusion criteria.

Thank you for your remarks. We expanded the introduction section about inflammation and its implications in neoplasia development as you suggested (l.44-54) and provided a better explanation of the aim of our study (l. 87-92). In the case of chronic kidney disease and cancer differentiation, urea level was found to be the primary indicator of renal function. In both the control group and the RCC group, regardless of histological features, the urea levels were observed in the adult reference range of 16.6- 48.5 mg/dl. We have addressed this issue in l. 99 and also provided more information about inclusion criteria in the methodology section (l. 272-274).

My opinion is that you should have brought more details about neoplasia history of each patients and about its associated comorbidities. In this matter, you should include a complete medical history, and then perform multivariate analysis between different conditions, neoplasia type and NA metabolites values.

A complete medical history of patients and the control group was added in Table 1. Because the most common comorbidity among ccRCC patients was hypertension, we performed a statistical analysis between ccRCC patients with and without a history of hypertension. After unpaired Student’s t-test, the NA metabolites were not statically different between the hypertensive and non-hypertensive group (p=0.645 for MNA; p=0.364 for NA; p=0.407 for Met4PY; p=0.466 for Met2PY; p=0.597 for nicotinic acid). We have captured this issue in l. 99-101. Given that other comorbidities do not constitute a high percentage of patients, multivariate analysis might not be statistically possible. We have also clarified that patients with co-occurrence of any non-renal cancer did not meet inclusion criteria (l. 274-276).

In your discussion section, you should talk more about the results from your study and not to include so much evidence from current medical literature and studies already published regarding this fields. I think these details should we written in introduction section, were you should explain more about the current evidence regarding this topic, bringing the latest highlights in this topic; thereafter, in your results focus only on your results and explain the best results from your study, and then explain what your study brings new in this field.

As the Reviewer suggested, we have modified the discussion section by moving some relevant issues to the introduction section (now in l. 79-86) and better explained what new our research brings to the field (l. 205-210).

In conclusion, please state only the novelty of your study, without pulling conclusion from current literature, because this section of your paper should be based only on paper originality. Also, you lack the limitations of your study and we can identify some majore ones like incomplete medical history of the patients and associated comorbidites, small cohort of patients. Please explain is your study brings something new in this field and how you can integrate your results into clinical practice, how your study results could be included maybe in prognostic scores or diagnostic tools.

To better emphasize the utility of our research we have significantly rephrased both the discussion and the conclusion sections. As suggested, we have added the limitations of our study (l. 244-252).

This topic is very interesting to discuss because we need a better management of neoplasia patients, given the fact that this is a condition with increased morbidity and mortality. We need better ways to asses this patients to increase treatment success.

Thank you for appreciation of our manuscript. We fully support this idea, therefore we have captured this statement in l. 257-259.

Round 2

Reviewer 1 Report

Comments and Suggestions for Authors

The manuscript has been improved and can be published.

Comments on the Quality of English Language

Only typos/minor editing

Reviewer 2 Report

Comments and Suggestions for Authors

Dear authors, 

Thank you for taking into consideration my advices regarding your manuscript. My opinion is that, after revision, your manuscript it is more appropiate for publication, providing a well structured and comprehensive manuscript. I consider that you took notice of my recomendations and now I think the your paper is suitable for publication. 

I wish you the best of luck with your future studies and publications. 

Great work!